# Estimating causes of community death of adults in Myanmar from a nationwide population sample: Application of verbal autopsy

Khin Sandar Bo[1‡], Sonja M. Firth[1‡], Tint Pa Pa Phyo[2], Nyo Nyo Mar[2], Ko Ko Zaw[3], Naw Hsah Kapaw[2], Tim Adair[1]*, Alan D. Lopez[4]

1 Melbourne School of Population and Global Health, The University of Melbourne, Carlton, Victoria, Australia, 2 Central Statistical Organization, Nay Pyi Taw, Myanmar, 3 University of Community Health, Magway, Myanmar, 4 Independent Consultant, Tamborine Mountain, QLD, Australia

‡ KSB and SMF contributed equally to this work and are joint first authors.
* timothy.adair@unimelb.edu.au

**Data Availability Statement:** The data used in this study are from the civil registration and vital statistics system of Myanmar. The data are owned

## Abstract

In Myanmar 84% of deaths occur in the community, of which half are unregistered and none have a reliable cause of death (COD) recorded. Since 2018, Myanmar has introduced improved registration practices and verbal autopsy (VA) to assess whether such methods can produce policy relevant information on community COD. Community health midwives and public health supervisors grade II collected VAs on over 80,000 deaths which occurred between January 2018 and December 2019 in a nationwide sample of 42 townships in Myanmar. Electronic methods were used to collect and consolidate data. The most probable COD was assigned using the SmartVA Analyze 2.0 computer algorithm. Completeness of VA death reporting increased to 71% in 2019. Most adult (12+ years) deaths (82%) were due to non-communicable diseases, primarily stroke, ischemic heart disease and chronic respiratory disease, for both men and women. VA results were consistent with Global Burden of Disease (GBD) Study estimates, except for cirrhosis in men, which was more common, and had a younger age distribution of death than the GBD. Large scale implementation of improved death registration practices and COD diagnosis using VA is feasible and provides plausible, timely, disaggregated and policy relevant information on the leading causes of community death. Addressing the burden of non-communicable diseases, particularly cirrhosis in young men, is an important public health priority in Myanmar. Improving completeness of VA death reporting in poorly performing townships and in neonates, children and women will further improve the policy utility of the VA data.

## Introduction

Myanmar, like many countries in Asia, has made substantial progress in improving child survival and reducing risks from infectious and communicable diseases, including malaria, tuberculosis and HIV/AIDS [1]. Non-communicable diseases (NCDs) have likely increased with

by the Government of Myanmar, who apply restrictions to the availability and use of these data. The data in this study were used with the permission of the Government of Myanmar. Anyone wishing to access this data would need permission from the civil registration and vital statistics system of the Government of Myanmar. Hence, we are unable to make the data fully available for this manuscript. To request permission from the Government of Myanmar, contact dgcso32.mopf@gmail.com or by writing a message in "Connect with us" at https://www.csostat.gov.mm/AboutCSO/OrganizationStructure.

**Funding:** This study was funded under an award from Bloomberg Philanthropies to the University of Melbourne and Vital Strategies to support the Data for Health Initiative. The Myanmar government provided in-kind support. The funders had no role in study design, data collection and analysis, decision to publish, or preparation of the manuscript.

**Competing interests:** The authors have declared that no competing interests exist.

development, but the extent, differential occurrence and speed of epidemiological transition in the country is difficult to determine given that an estimated 84% of all deaths occur in the community without an adequate cause of death (COD) being assigned [2]. Further, around a half of these deaths are not even registered or reported to the Central Statistical Organisation (CSO) [3]. Health policy and planning in Myanmar thus relies on information from less than one-fifth of all deaths that occur in hospital, information that is not representative of the wider population and is itself of low quality [3]. Reliable, representative and complete COD information is essential to guide health policies and the evaluation of intervention impact to improve population health, as well as for monitoring progress with the Sustainable Development Goals (SDGs), over 40% (7/17) of which require cause-specific mortality data [4].

In recognition of the key role of timely, complete and good quality COD statistics in guiding development strategies, the government of Myanmar has, since 2016, partnered with the Bloomberg Philanthropies Data for Health Initiative (D4H) [5], to undertake a series of civil registration and vital statistics (CRVS) strengthening activities. A key component of this strategy involved the introduction of the routine use of automated verbal autopsy (VA) methods, to improve knowledge about levels, patterns and causes of mortality for the vast majority of the population dying outside of a health facility. In 2017 these methods were piloted in 14 townships in two states and one region of Myanmar, representing around 3% of the national population. Following an independent evaluation [6], these methods were subsequently extended to 42 townships nationwide.

In this paper we report the findings of over 80,000 VA's collected in this nationwide sample during 2018 and 2019, demonstrating the utility and feasibility of using routine application of automated VA methods to improve completeness of COD reporting and hence measure community COD patterns among adults aged 12 years and over, thereby filling a critical data gap in the country's health information.

## Methods

### Site selection and sampling strategy

As part of the D4H, improved registration practices and routine automated VA were implemented for community deaths in a sample of 42 townships in the country (see Fig 1). There were 14 townships within three regions/states (Sagaing region, Magway region and Mon state) included in a pilot of the routine automated VA that, for practical reasons, were also included in the sample of townships. For the remaining states and regions and NayPyiTaw (NPT) national territory, two townships were chosen, with Shan state divided (due to its size) into three and two townships selected from each. The selection of townships was based on those from a National Survey on causes of death in Myanmar using verbal autopsy conducted in 2017 [7] with half of the final selection of townships matching this sample. Consideration was also given to the feasibility of implementation and the willingness of the township health department to collect VA on a routine basis. These 42 townships (out of a total of 330 in the country) covered a total population of 8.4 million, representing about 15% of the national population. The population of each township was estimated using the 2014 Census population projections from the Department of Population. (S1 Table). The 42 townships in the sample are representative of the epidemiological and demographic characteristics of the national population, in terms of the population age structure (6.1% were aged 65 and over, compared with 6.4% nationally) and the estimated under-five mortality rate (45.4 per 1000 compared with 44.6 per 1000 nationally; see S1 Text). Data collection in the 14 pilot townships commenced in January 2017 and concluded in March 2020, while in the remaining 28 townships it commenced in February 2018 and concluded in March 2020. In each township, VA was conducted

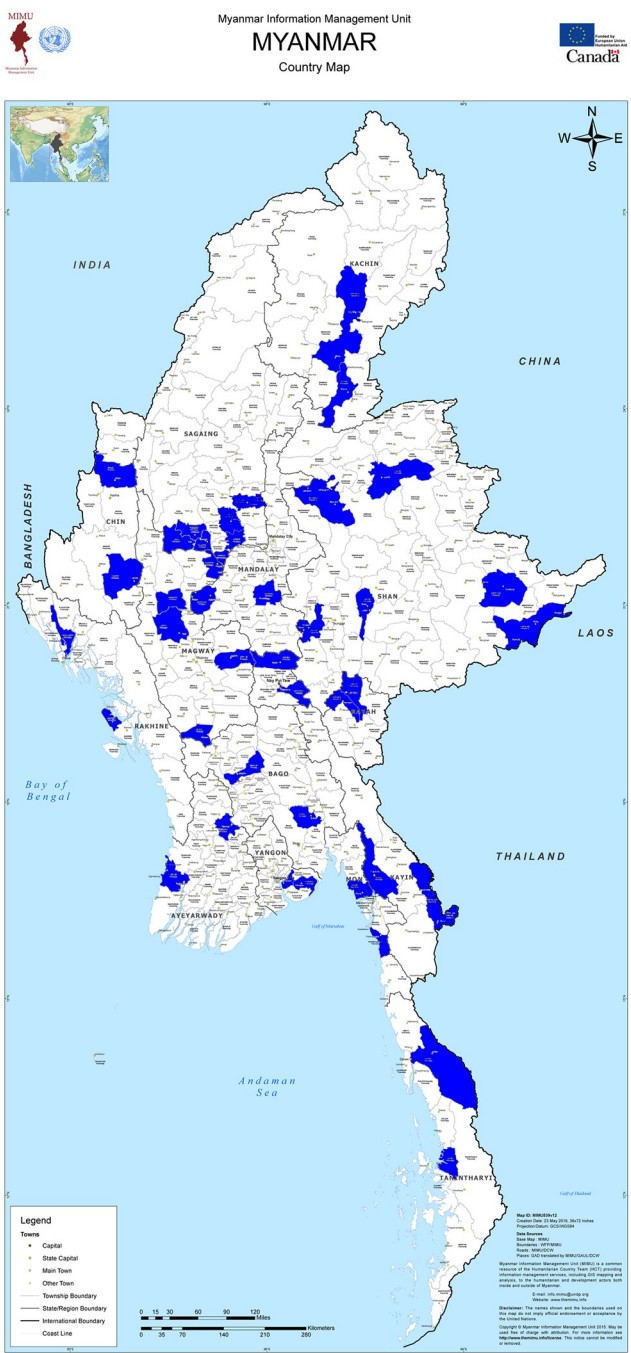

**Fig 1. Geographical spread of townships implementing verbal autopsy in Myanmar.** Source: Myanmar Information and Management Unit. http://themimu.info/.

on all notified community (out of hospital) deaths that occurred between January 2018 and December 2019 as part of routine death registration procedures in these sites.

## Verbal autopsy tool and automated methods for assigning cause of death

Automated VA is the only feasible alternative to medical certification of COD for populations with limited physician density. The method involves an interview with a family member or

close associate of the deceased to collect information on signs and symptoms experienced by the deceased prior to death using a tablet, and assignment of the probable COD using an automated computer algorithm, which has been shown to be as accurate as physicians in assigning a COD from VA [8]. The primary output of VA is a set of cause specific mortality fractions (CSMF) for the population according to three broad age-groups, Adult, Child and Neonate. WHO VA standards dictate the age-ranges for these broad age-groups, whereby 'Adult' refers to deaths in people aged 12 and over, children constitute ages 29 days to 11 years and neonates 0–28 days [9].

The Myanmar language version of the Population Health Metrics Research Consortium (PHMRC)-shortened verbal autopsy questionnaire [10], loaded into the ODK collect application on a tablet [11], was used to collect VA information with completed interviews transferred to the ODK aggregate website at the CSO. CODs were assigned by the SmartVA Analyze application 2.0.0 [12], which assigns the most probable COD from a target cause list of 33 adult causes, 21 child causes and six neonatal causes, based on the Tariff 2.0 diagnostic method [13, 14]. Tariff 2.0 produces two main outputs; an individual COD prediction file and an aggregated VA population CSMF file. In the individual prediction file, deaths with insufficient information for SmartVA Analyze to reliably predict a COD were assigned an "undetermined" COD. At the population level (represented by the CSMF), undetermined CODs are redistributed among the SmartVA causes, based on the likelihood of specific causes being misdiagnosed as such, and on the estimated cause pattern for the population from the GBD, as described elsewhere [14]. The SmartVA questionnaire and Tariff automated method to assign COD (together referred to as SmartVA) have been validated using a gold standard dataset of hospital deaths on which VA was also performed [15–18]. This method has been applied in several countries in order to better understand the causes of community deaths [19].

## VA Interviewers, training and supervision

VA interviewers comprised basic health staff, including midwives and public health supervisors grade II (PHSII), a new cadre who worked under the midwives at some sub-rural health units. Midwives are tasked with registration of deaths, however due to their competing commitments and the fact that they collect this data both for the CSO (the agency tasked with compiling vital statistics) and for the health information system, completeness of death registration for community deaths has been low. A five day VA curriculum [20] was delivered to 3,965 basic health staff, covering all areas of death registration and VA including the importance of death registration, the uses of COD information, comprehension of the different modules in the questionnaire, how to use the tablet to collect information, how to deal sensitively with the family of the deceased, ethics, practicalities of collecting VA data, and supervisory support. 650 supervisors were also trained in VA methods and supported the VA interviewers, attending some interviews with them, as well as checking the administrative data from the VA interview prior to transmitting the forms to the ODK aggregate server. In addition, the township CSO monitored death registration forms (Form 201) and VA monitoring forms to ensure VA was conducted for all deaths registered. Evaluation workshops were conducted in townships and provided an opportunity to provide feedback to these frontline workers, address their challenges and concerns and coach them on how to deliver the VA intervention more effectively. Further information on engagement with VA interviewers, training and supervisory activities can be found elsewhere [21].

## Data quality checks and data analysis

Prior to analysis, the following checks were performed on the VA data by staff from the CSO with anomalies clarified with the township medical officer.:

- removal of duplicate records

- availability of age and sex and code of the areas in which VA was collected

- total number of VA matched with the monthly summary reports provided through the township medical officer (TMO).

- consistency between age and COD, and between sex and COD.

In addition, all maternal deaths were subjected to a more detailed follow up with the results of this investigation being sent to the state/region Department of Public Health.

The VA output data were interrogated, using the Verbal Autopsy Interpretation and Performance Evaluation Resource (VIPER) [22], and associated guidelines [23]. These resources propose four steps to improve understanding and interpretation of the mortality data from VA: i) assessment of the characteristics of the VA population, ii) completeness of the VA death reporting, iii) plausibility of the age-sex distribution of deaths from VA, and iv) plausibility of the CSMF from VA.

Completeness of VA death reporting (the % of estimated community deaths with a VA) was calculated using the empirical completeness method, which uses data inputs that reflect the key demographic drivers of the level of the crude death rate in the VA population, namely; the percentage of the population aged 65 years and above, and the under-five mortality rate [24]. (See S1 Text)

Global Burden of Disease (GBD) estimates were used as comparator data for this analysis. The age-sex distribution of VA deaths for various causes were compared with the corresponding cause-specific distributions from the GBD Study, estimated from the extensive GBD dataset on causes of death across countries and over seven decades, and various time series of data on key covariates that predict causes of death [25].

Although VA was collected on all deaths, we limited the COD analysis to adult deaths (ages 12 and over) because deaths in children and neonates were under-represented in our sample (3% compared to 11% according to the GBD comparator data), and so their COD results would be biased. Further, given the very similar age profiles of deaths in 2018 and 2019, COD analyses have been conducted on combined data, unless otherwise stated. The age-distribution of the sample population aged 12+ years can be found in S2 Table.

## Ethical considerations

As part of standard operating procedures for VA data collection, informed verbal consent was obtained from the relatives of the deceased person, and the Ministry of Health and Sports (MOHS) and CSO approved this consent procedure for the VA interviews in Myanmar. Verbal consent was obtained from the parents or guardian for respondents aged under 18 years. Participation was voluntary, and participants had the right to refuse or withdraw at any time during data collection. The entire data set was kept secure and confidential at the CSO and analysis was conducted on de-identified data. Emotional support was offered to informants, as required. Appropriate care was taken to ensure that participants were not put at any risk; none received compensation for taking part in this data collection.

## Results

The vast majority (over 97%) of VAs collected were adult deaths, with just over half (56%) being male deaths (Table 1). In total, 82,401 VAs were carried out on deaths that occurred between January 2018 and December 2019. Male deaths occur at younger adult ages compared to females, with deaths at 80+ years in females almost twice that of males (Fig 2). Males also

**Table 1. Number (%) of VAs by sex and age (adult, child and neonate), 2018–2019.**

| Age group | 2018 | | | 2019 | | |
|---|---|---|---|---|---|---|
| | Male (%) | Female (%) | Both sexes (%) | Male (%) | Female (%) | Both sexes (%) |
| Adult | 22,007(54·6) | 17,211 (42·7) | 39,218 (97·2) | 23,167 (55·1) | 17,781(42·3) | 40,948(97·3) |
| Child | 510 (1·3) | 340 (0·8) | 850 (2·1) | 484 (1·2) | 358 (0·9) | 842 (2·0) |
| Neonate | 170 (0·4) | 101 (0·3) | 271 (0·7) | 152 (0·4) | 120 (0·3) | 272 (0·6) |
| Total | 22,687 (56·2) | 17,652 (43·8) | 40,339 (100) | 23,803 (56·6) | 18,259 (43·4) | 42,062 (100) |

died at younger adult ages in the VA population compared to the GBD, particularly at ages 30–49 years. (S1 Fig).

## Completeness of VA reporting

Overall, in the first year of VA implementation in the 42 townships in 2018 completeness of VA reporting was 67% of estimated community deaths and increased slightly to 71% in 2019 (Table 2). There was a marked sex differential in completeness in favour of males, increasing slightly from 16 to 18 percentage points over the two-year period of data collection. Completeness of reporting varied considerably across the 42 townships, ranging from 17% to over 90%, exceeding 65% in two-thirds of all townships in 2019. (S3 Table).

## Leading causes of death

The broad COD structure, defined by GBD Level 1 cause groupings, in Myanmar identified by the VA data was very similar to that estimated by the GBD, suggesting that the VA data are

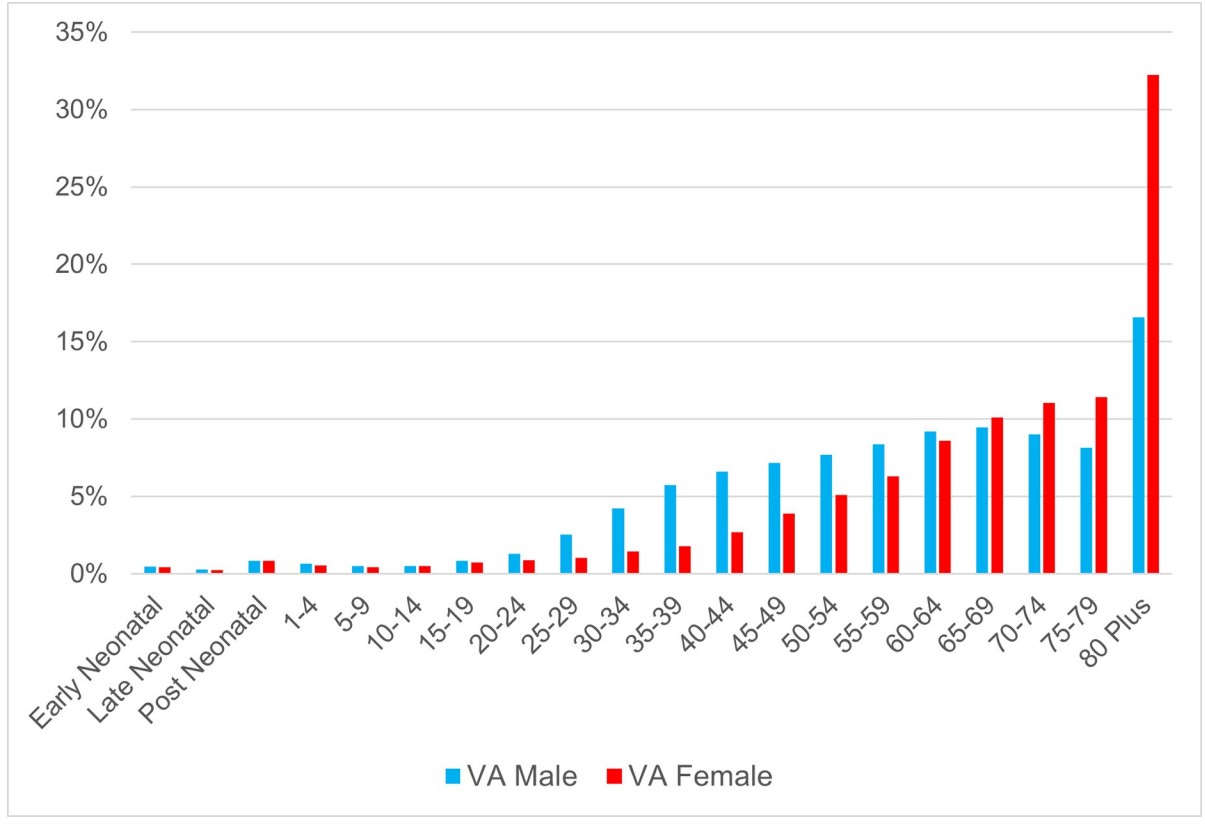

**Fig 2. Age-sex distribution of VA in 42 townships, Myanmar (2018–2019).**

**Table 2.  Estimated VA completeness as a proportion of community deaths, 42 townships, 2018 and 2019.**

| 2018 | | | 2019 | | |
|------|------|------|------|------|------|
| Male | Female | Total | Male | Female | Total |
| 75·7% | 59·6% | 66·6% | 80·7% | 62·4% | 70·7% |

consistent with other assessments of the country's stage of epidemiological transition (Table 3).

The leading causes of death suggested by VA are shown in Table 4 for males and females separately. These data emphasise the dominance of NCDs, accounting for the top six and top five causes for males and females respectively. Cirrhosis deaths are comparatively more prominent in men (14%) than women (3%) as are lung cancer, AIDS, and road traffic accidents which feature in the top 15 male causes but not for females. Ischemic Heart Disease (IHD) has a higher CSMF and rank in females compared to males (16% and 10% respectively). In both datasets, the top 15 causes of death comprise around 90% of all deaths. A comparison between the VA data and the GBD shows a close agreement between the two data sources for the leading COD in both males and females. A notable exception is liver cirrhosis for males which is substantially more important as a COD in the VA data (14%) compared to the GBD (8%). (S4 Table).

The leading causes of death for men within broad age groups are shown in Fig 3A–3C. AIDS, road traffic accidents, tuberculosis and other infectious diseases appear among the leading COD in the younger adult male age-group (less than 50 years old). More surprisingly, NCDs already dominate at these ages, with cirrhosis, stroke, IHD and other NCDs causing about half of all deaths in men under age 50. At older ages, stroke, IHD, chronic respiratory diseases and diabetes account for almost two-thirds of deaths, with pneumonia and TB among the top 10 causes as well.

Similar data for women are shown in Fig 4A–4C. As for male deaths, the top causes of death under age 50 for women are dominated by NCDs, accounting for the top six causes (and 56% of all deaths) in this age group. Maternal causes, AIDS, cervical and breast cancer each account for about 5% of deaths in women of reproductive age, while stroke, IHD, chronic respiratory diseases and diabetes collectively account for about 60% of deaths in women beyond age 50.

An additional plausibility check on the VA COD findings is to assess the age distribution of deaths for leading causes, which should follow a predictable pattern, as identified in the comparator GBD data. With very few exceptions, the age distribution of the leading 15 causes of death from VA closely aligns with that estimated in the GBD. (S2 Fig). Notable exceptions, particularly for males, include deaths from cirrhosis, chronic kidney disease, AIDS, and to a lesser extent, leukemia/lymphoma and lung cancer, all of which appear to have a slightly younger age distribution of death than the GBD suggests.

## Regional variations in CSMF

A regional analysis of CSMFs suggests that this NCD-dominant cause pattern is consistent across sites in Myanmar (Table 5), with some exceptions, notably Kachin, which has a higher

**Table 3.  Adult death by broad cause group, VA and the GBD [26].**

| Broad cause group (Level 1 causes GBD) | VA (2018 and 2019) | GBD 2019 |
|------|------|------|
| Group I– Communicable, maternal, neonatal and nutritional diseases | 12·4 | 12·7 |
| Group II– Non-communicable diseases | 81·7 | 81·7 |
| Group III– Injuries | 5·9 | 5·6 |

**Table 4. Cause specific mortality fractions (CSMFs) (%), leading 15 causes of death in adult males and females (VA 2018/2019).**

| Males | % | Females | % |
|---|---|---|---|
| Stroke | 24 | Stroke | 25 |
| Cirrhosis | 14 | Ischemic Heart Disease | 16 |
| Chronic Respiratory | 10 | Chronic Respiratory | 12 |
| Ischemic Heart Disease | 10 | Diabetes | 8 |
| Diabetes | 5 | Other Non-communicable Diseases | 4 |
| Other Non-communicable Diseases | 5 | Pneumonia | 4 |
| Pneumonia | 4 | Chronic Kidney Disease | 3 |
| Tuberculosis | 3 | Cirrhosis | 3 |
| Chronic Kidney Disease | 3 | Cervical Cancer | 2 |
| AIDS | 2 | Breast Cancer | 2 |
| Prostate Cancer | 2 | Leukemia/Lymphomas | 2 |
| Leukemia/Lymphomas | 2 | Tuberculosis | 2 |
| Lung Cancer | 2 | Diarrhoea/Dysentery | 2 |
| Esophageal Cancer | 2 | Esophageal Cancer | 1 |
| Road Traffic | 2 | Other cancers | 1 |

fraction of deaths attributed to AIDS than elsewhere. The top five causes of death (both sexes) nationwide–stroke, IHD, chronic respiratory disease, cirrhosis and diabetes—appear as the top five causes in 12 of the 15 states and regions. In the three other states/regions, at least three of these causes appear in the top five and all feature in the top 10. Stroke was the leading COD in all states/regions.

## Undetermined cause of death

Deaths for which the cause could not be determined by VA ('undetermined' deaths) are more common at older ages. (S3 Fig). There is a marked decrease in the level of cause-undetermined deaths for both males and females over the course of the data collection (Table 6). Undetermined levels varied by state and region, ranging from 11 to 23 per cent for the 2018 and 2019 VA data combined. (S5 Table).

## Discussion

This is the first nationwide empirical analysis of community COD patterns in Myanmar using routinely collected VA data. Our findings suggest that VA is a practical alternative to medical certification to generate policy-relevant information on the leading causes of death for the approximately 84% of deaths in the country that occur in the community. The geographic spread and the similarity of the population age structure between the VA sites and the country, support the belief that the findings represent mortality conditions for the Myanmar national population. Disaggregated data provided to townships on a routine basis can also be used to monitor cause of death trends in a timely manner and support local health planning and service improvement. Community death data from the 2018 and 2019 VA data collection has for the first time been published in the Myanmar Statistical Yearbook 2020 [27]. Mortality and COD data from VA has also been integrated with hospital data (COD ascertained by doctors using a medical certificate of cause of death) from the 42 townships to provide national-level COD estimates which have been used to calculate baseline values for monitoring national progress with the NCD-related SDGs [28]. VA will likely be required for many years in Myanmar until complete medical certification of community deaths is possible.

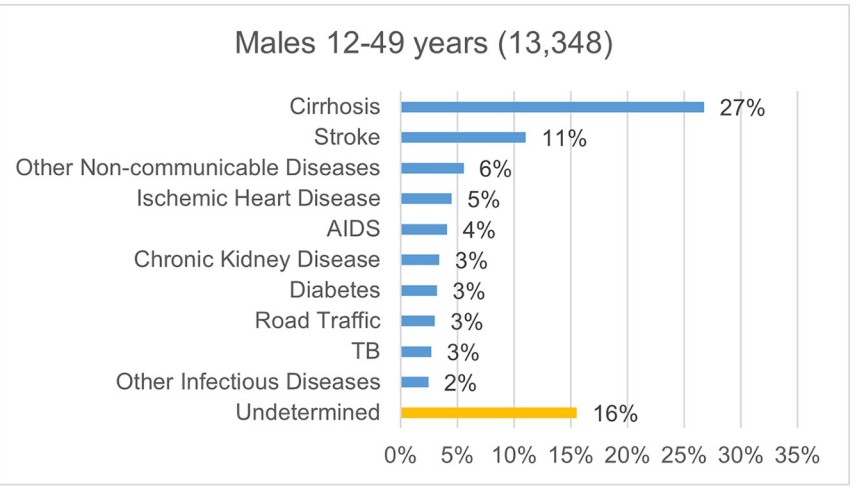

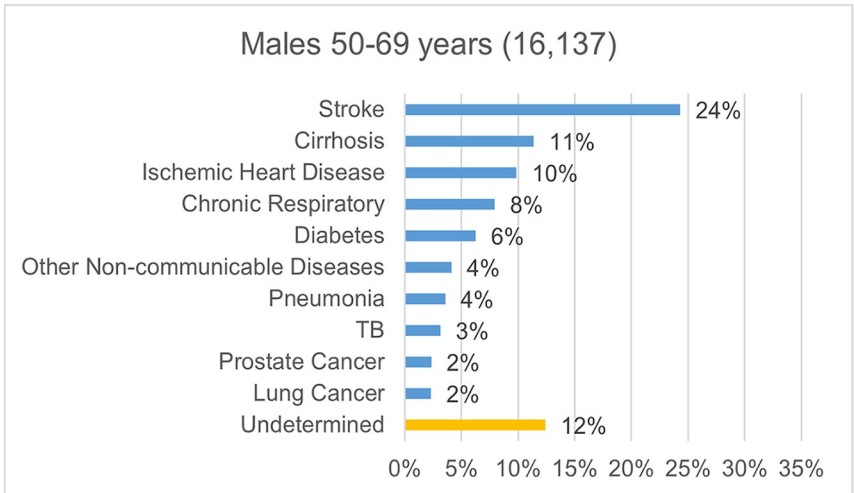

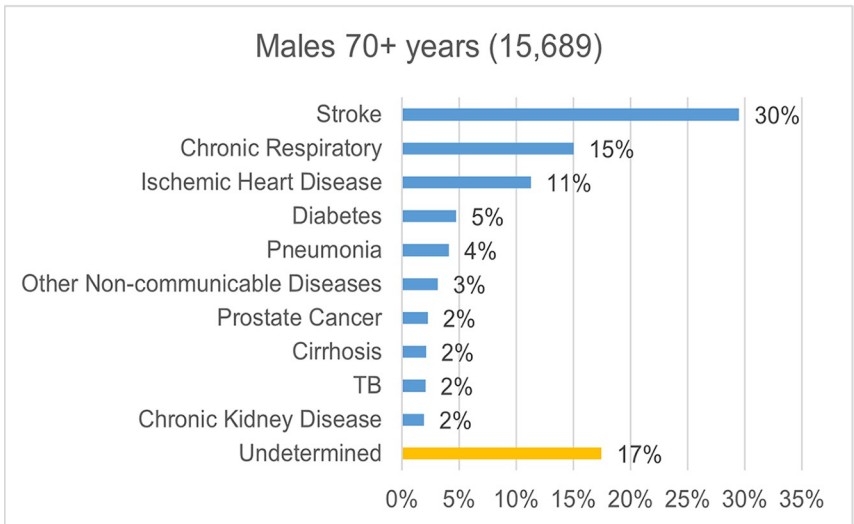

**Fig 3. A-C.** Top 10 causes of death in men at younger, middle and old ages, 42 townships in Myanmar, 2018/2019 (Tariff individual prediction file).

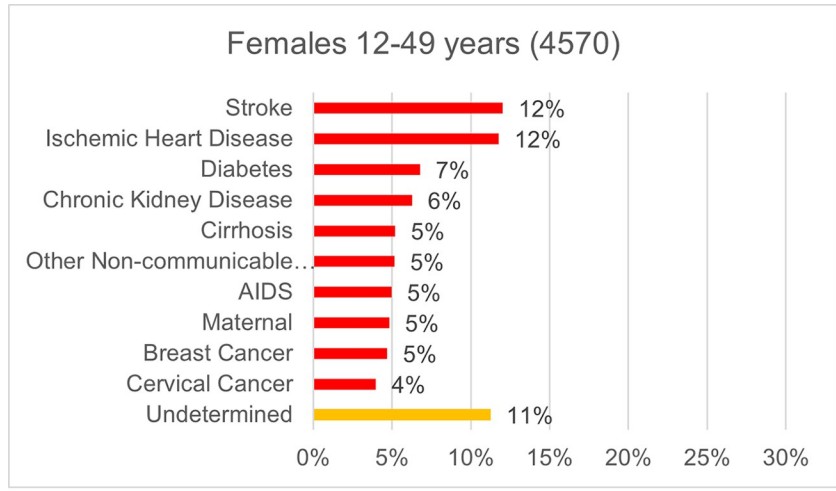

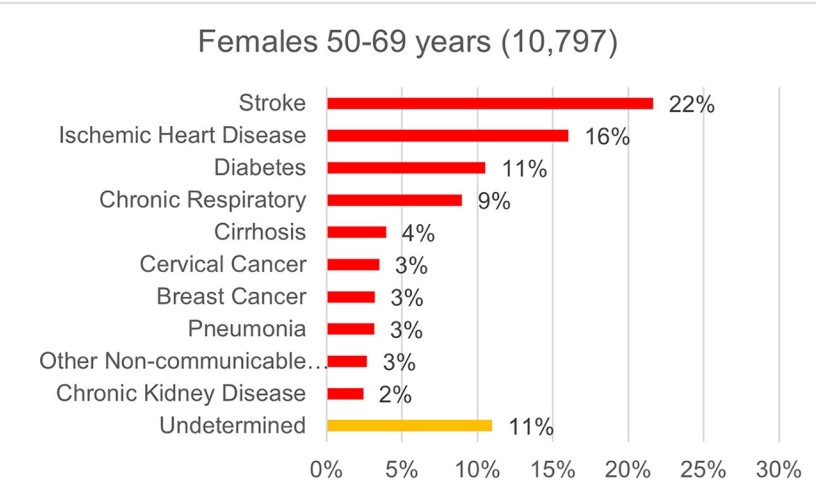

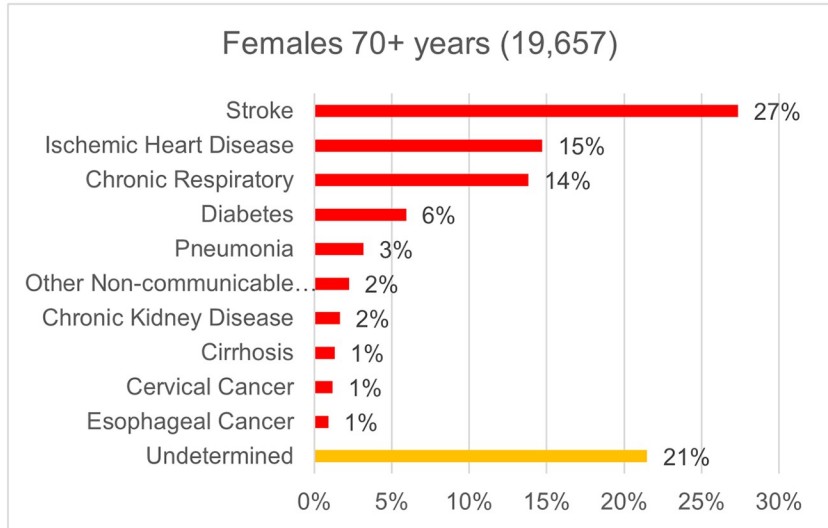

**Fig 4. A-C.** Top 10 causes of death in women at younger, middle and old age, 42 townships in Myanmar, 2018/2019 (Tariff individual prediction file).

**Table 5. CSMF of top 20 causes of death (Age 12+) by state/region (both sexes).**

| | Ayeyarwady | Bago | Chin | Kachin | Kayah | Kayin | Magway | Mandalay | Mon | NPT | Rakhine | Sagaing | Shan | Tanintharyi | Yangon |
|---|---|---|---|---|---|---|---|---|---|---|---|---|---|---|---|
| **No. of VAs 2018/ 2019 combined** | 6,489 | 4,627 | 750 | 3,856 | 1,546 | 7,092 | 8,691 | 6,540 | 5,099 | 3,585 | 2192 | 12,576 | 8,711 | 4,468 | 3,944 |
| **Top 15 Causes** | | | | | | | | | | | | | | | |
| **Stroke** | 24% | 23% | 14% | 20% | 21% | 21% | 28% | 28% | 26% | 20% | 29% | 26% | 20% | 25% | 24% |
| **Ischemic Heart Disease** | 13% | 15% | 9% | 10% | 9% | 11% | 9% | 11% | 16% | 14% | 5% | 12% | 13% | 13% | 20% |
| **Chronic Respiratory** | 12% | 11% | 12% | 6% | 7% | 9% | 13% | 13% | 9% | 12% | 9% | 11% | 10% | 14% | 7% |
| **Cirrhosis** | 7% | 8% | 7% | 11% | 12% | 9% | 8% | 10% | 9% | 10% | 4% | 10% | 9% | 7% | 11% |
| **Diabetes** | 7% | 5% | 5% | 7% | 5% | 6% | 6% | 5% | 7% | 8% | 5% | 6% | 8% | 7% | 7% |
| **Other NCDs** | 5% | 4% | 12% | 4% | 8% | 6% | 5% | 4% | 6% | 4% | 9% | 5% | 5% | 4% | 3% |
| **Pneumonia** | 3% | 4% | 7% | 5% | 5% | 5% | 4% | 5% | 3% | 5% | 3% | 4% | 4% | 4% | 4% |
| **Chronic Kidney Disease** | 3% | 3% | 4% | 4% | 4% | 3% | 3% | 3% | 3% | 4% | 3% | 3% | 3% | 2% | 4% |
| **TB** | 3% | 3% | 2% | 4% | 3% | 4% | 2% | 2% | 2% | 3% | 3% | 2% | 3% | 2% | 2% |
| **Leukemia/ Lymphomas** | 2% | 2% | 2% | 1% | 3% | 2% | 2% | 2% | 2% | 2% | 2% | 2% | 2% | 2% | 2% |
| **AIDS** | 2% | 1% | 1% | 6% | 1% | 1% | 1% | 1% | 2% | 2% | 1% | 1% | 2% | 2% | 1% |
| **Diarrhea/Dysentery** | 1% | 1% | 2% | 2% | 2% | 1% | 2% | 1% | 1% | 2% | 2% | 2% | 2% | 1% | 1% |
| **Lung Cancer** | 2% | 2% | 1% | 1% | 1% | 2% | 1% | 1% | 2% | 1% | 1% | 2% | 1% | 2% | 2% |
| **Esophageal Cancer** | 2% | 2% | 1% | 1% | 2% | 2% | 1% | 1% | 2% | 1% | 2% | 1% | 1% | 2% | 2% |
| **Falls** | 1% | 1% | 4% | 2% | 2% | 1% | 2% | 1% | 1% | 1% | 2% | 1% | 2% | 1% | 0% |
| **Malaria** | 1% | 1% | 1% | 1% | 1% | 1% | 1% | 1% | 1% | 2% | 2% | 1% | 1% | 1% | 1% |
| **Prostate Cancer** | 1% | 1% | 2% | 1% | 1% | 1% | 1% | 1% | 1% | 1% | 1% | 1% | 1% | 1% | 1% |
| **Cervical Cancer** | 1% | 1% | 1% | 1% | 1% | 1% | 1% | 1% | 1% | 1% | 1% | 1% | 1% | 1% | 1% |
| **Other Infectious Diseases** | 1% | 1% | 1% | 1% | 2% | 1% | 1% | 1% | 1% | 1% | 1% | 1% | 1% | 1% | 1% |
| **Breast Cancer** | 1% | 1% | 1% | 1% | 1% | 1% | 1% | 1% | 1% | 1% | 1% | 1% | 1% | 1% | 1% |

VAs were conducted on the majority of community deaths (71% in 2019) estimated to have occurred in the 42 townships, providing some confidence that the results broadly reflect community COD patterns. While the overall completeness of VA death reporting improved, probably due to the intervention becoming more established throughout the data collection period, there was substantial variation amongst the VA sites. Poor VA completeness in some sites can be explained by high staff turnover, conflict and access issues, noted previously [3]. There is also considerable under-reporting of neonatal and child death. While there may be a greater likelihood that children, as opposed to adults, would be taken to hospital and might die there, cultural factors undoubtedly affect more complete reporting of child and neonatal deaths [3]. The large sex differential in the reporting of community deaths, favouring males, may be partly due to the empirical completeness method over-estimating male completeness because of high

**Table 6. Percent undetermined cause of death among adult, by six monthly intervals, January 2018 to December 2019, 42 townships.**

| | Jan-Jun 2018 | Jul-Dec 2018 | Jan-Jun 2019 | Jul-Dec 2019 |
|---|---|---|---|---|
| Both sexes | 20·0 | 15·9 | 14·4 | 13·3 |
| Male | 18·9 | 15·1 | 13·9 | 12·5 |
| Female | 21·5 | 17·0 | 15·0 | 14.3 |

male adult mortality when compared to child mortality, the latter being an input into the method. Further investigation is needed to assess the extent and the principal reasons.

The VA results suggest that the epidemiological transition in Myanmar is well advanced, with more than 8 in 10 adult deaths attributable to NCDs, a finding that is consistent across all states and regions. This has clear policy implications, including a greater focus on controlling hypertension and on reducing prevalence of smoking and drinking particularly in young men, who die prematurely from diseases attributable to these behaviours, including stroke, IHD, chronic respiratory diseases and liver cirrhosis [29, 30]. Moreover, the close alignment between our estimates of leading COD and those from the GBD provides some basis for confidence in our results. Whilst there is considerable uncertainty in the GBD estimates for countries like Myanmar, due to lack of data, the GBD findings nonetheless represent the most probable COD patterns that are associated with prevailing levels of key health determinants, as measured by covariates, based on modelling of decades of epidemiological and demographic information [25]. The VA results are also consistent with those from a cross–sectional National Survey on Causes of Death in Myanmar completed in 2017, primarily conducted to assess mortality due to HIV/AIDS, Tuberculosis and Malaria, which employed similar VA methods [7]. Road traffic accidents were a more prominent COD in the national survey dataset (as well as the GBD estimates). Both datasets, unlike VA, include hospital deaths which are more likely than community deaths to capture acute road traffic accidents [7]. While the CSMF from hospitals is different to that in the community, only one-in-six deaths occurs in hospital, as demonstrated in the integration of VA and MCCOD data the population CSMF follows that of community deaths [28]. The variation in COD patterns observed between states and regions reflects local epidemiological factors. Hence, the higher proportionate mortality from AIDS in Kachin compared to the rest of the country likely reflects the high levels of drug use and presence of sex workers associated with the jade mining industry there [31].

For prevention policy, particularly to reduce premature deaths, it is important to understand the leading causes of death at different ages, and by sex. The younger age distribution of deaths among adult males (age 30–49 years) compared to international estimates has been reported previously [3], and is likely to be real. The younger age-pattern of cirrhosis deaths in men than predicted by the GBD suggests that this is a more important public health challenge in Myanmar than previously thought and that public policy action to reduce this exposure is urgent. Myanmar has high levels of alcohol consumption in males, starting at young ages [32], and a high prevalence of Hepatitis B and C [33]. Strengthening policies to reduce alcohol use and removing barriers to the treatment of Hepatitis B and C [34], would appear to be important public health priority actions for government to reduce excess mortality due to this disease, particularly in men of economically productive age.

The fraction of deaths for which SmartVA could not predict a diagnosis (<15% in 2019) is not unusual for automated diagnostic methods applied to community deaths [19]. The slightly higher proportion among females largely reflects their older age at death. The likelihood that an automated diagnostic algorithm can reliably predict a specific cause depends strongly on the skill, training, and motivation of interviewers in extracting appropriate information from respondents. To the extent that these skills increase with experience, it might be expected that the fraction of undetermined cases would decrease over time, as demonstrated in this analysis. During the scale-up of VA, considerable emphasis was placed on consultation with frontline workers, ensuring they were properly trained, supported and informed of the results of their efforts.

## Limitations

The low levels of child and neonatal VAs recorded in our data collection makes it difficult to conduct a meaningful analysis of the CSMFs for these age-groups, hence we did not perform

this analysis. Greater efforts are needed to understand and overcome barriers to improved completeness of VA death reporting in these age-groups to enable the results of VA to inform child survival strategies.

The townships where VA data were collected are subject to some selection bias. Townships in more remote areas, with sparser populations, as well as those more likely to be affected by conflict (such as some townships in Rakhine and Shan states), were excluded for operational reasons. In addition, the low number of VAs and low completeness of VA in some townships will affect the representativeness of the data for these areas and thus the generalisability of the results to the national population. If the COD patterns in townships with low completeness are different from other townships, then this may might bias the national-level results. However, because the COD pattern is consistent across our sites, we believe the bias to be minimal.

Although covering the major causes of public health importance, VA is constrained in terms of the number of causes of death that it can generate compared to a medical certificate of COD. This limitation can result in a relatively high proportion of 'residual' causes where there is enough information to specify the 'type' of cause (e.g. Cancer) but not enough to define a specific cause (such as primary cancer site). In Myanmar, "Other NCDs" make up a relatively high proportion of all VAs (5%). To improve the policy utility of the VA data, further investigation of the relative frequency of specific causes from hospital data may help to understand the relative importance of various causes that comprise this residual category.

Finally, although our analysis was based on over 80,000 VAs, some CSMFs were based on small numbers of deaths and are therefore subject to uncertainty due to small numbers. This is more of a concern for specific age groups and at the state/region level. It is not likely to affect the ranking of the top five to eight causes, which provide compelling evidence for public health action to avoid the major causes of premature deaths in the country.

## Conclusions

Our findings suggest that the implementation of improved death registration practices and routine application of automated VA in Myanmar can yield plausible and critical information on the causes of community deaths in the country. Myanmar showed significant progress in implementing and institutionalising improved death reporting and VA practices at scale between 2016–2020 [21]. The findings suggest that improving completeness of VA reporting in poorly performing townships, as well as among women, is needed to improve the policy utility, and generalisability, of these results. In addition, increasing completeness of VA reporting for children will allow similar analysis to be conducted and results for this age group to be used for health planning and policy. Nonetheless, the findings reported here on the initial application of the method provide compelling evidence of the key public health challenges facing Myanmar to reduce premature deaths and improve population health.

## Supporting information

**S1 Text. Estimation of the under-five mortality rate as input into the empirical completeness method.**
(DOCX)

**S1 Table. Projected mid-year population in Myanmar in the 42 townships 2018 and 2019.**
(DOCX)

**S2 Table. Population age distribution (age 12+) in 42 townships by sex.**
(DOCX)

**S3 Table. Completeness of verbal autopsy as a proportion of community deaths (by township).**
(DOCX)

**S4 Table. CSMFs (%) leading causes of death in adults VA compared to GBD 2019.**
(DOCX)

**S5 Table. Undetermined cause of death (adult) by state/region (2018/2019 VA data combined).**
(DOCX)

**S1 Fig. Age- sex distribution of death, 2018–2019 verbal autopsies in 42 townships and GBD 2019.**
(DOCX)

**S2 Fig. Age-distribution of top 15 causes—verbal autopsy compared to GBD2019.**
(PDF)

**S3 Fig. Age distribution of undetermined cause of death (2018/2019 VA data combined).**
(PDF)

## Acknowledgments

We would like to acknowledge the Myanmar government, in particular, national staff at the Central Statistical Organisation (CSO), Ministry of Planning, Finance and Industry (MOPFI) and from the Health Management Information Division, Ministry of Health and Sports as well as regional and peripheral health staff, Basic Health Staff, township CSO and staff from the General Administration Department working in the sites where VA has been implemented. Thanks to the mortality technical working group for providing leadership and guidance in the VA implementation process jointly supported by the Bloomberg Philanthropies Data for Health Initiative.

## Author Contributions

**Conceptualization:** Khin Sandar Bo, Sonja M. Firth, Ko Ko Zaw, Alan D. Lopez.

**Data curation:** Khin Sandar Bo, Sonja M. Firth.

**Formal analysis:** Khin Sandar Bo, Sonja M. Firth, Tint Pa Pa Phyo, Nyo Nyo Mar, Naw Hsah Kapaw, Tim Adair, Alan D. Lopez.

**Investigation:** Khin Sandar Bo, Sonja M. Firth, Ko Ko Zaw.

**Methodology:** Tint Pa Pa Phyo, Nyo Nyo Mar, Tim Adair.

**Project administration:** Khin Sandar Bo, Sonja M. Firth.

**Writing – original draft:** Khin Sandar Bo, Sonja M. Firth, Tim Adair, Alan D. Lopez.

**Writing – review & editing:** Khin Sandar Bo, Sonja M. Firth, Tim Adair, Alan D. Lopez.

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
