## [Decision Letter · Decision Letter 0]

24 Aug 2023

PGPH-D-23-01258

Estimating causes of community death of adults in Myanmar from a nationwide population sample: application of verbal autopsy

Dear Adair,

Thank you for submitting your manuscript to PLOS Global Public Health. After careful consideration, we feel that it has merit but does not fully meet PLOS Global Public Health’s publication criteria as it currently stands. Therefore, we invite you to submit a revised version of the manuscript that addresses the points raised during the review process.

We look forward to receiving your revised manuscript.

Kind regards,

Collins Otieno Asweto, PhD

Academic Editor

Journal Requirements:

Additional Editor Comments (if provided):

Reviewers' comments:

Reviewer's Responses to Questions

**Comments to the Author**

1. Does this manuscript meet PLOS Global Public Health’s publication criteria? Is the manuscript technically sound, and do the data support the conclusions? The manuscript must describe methodologically and ethically rigorous research with conclusions that are appropriately drawn based on the data presented.

Reviewer #1: Yes

Reviewer #2: Yes

2. Has the statistical analysis been performed appropriately and rigorously?

Reviewer #1: Yes

Reviewer #2: N/A

3. Have the authors made all data underlying the findings in their manuscript fully available (please refer to the Data Availability Statement at the start of the manuscript PDF file)?

Reviewer #1: No

Reviewer #2: Yes

4. Is the manuscript presented in an intelligible fashion and written in standard English?

Reviewer #1: Yes

Reviewer #2: Yes

5. Review Comments to the Author

Reviewer #1: • Data availability: The framing of the text that data cannot be fully made available means independent review of this data in the future is not possible. I would suggest that, it should be rephrased that the VA data is owned by Government and anyone wishing to access this data would need permission from the civil registration and vital statistics systems of the Government of Myanmar.

• Conclusion: The focus of this study is on adults (12years+ by definition), why would VA data of neonates and children further improve policy utilization of VA data? What data from this research supports this position/recommendation?

• Line 46: Note that 7/17 is not half: either use a little over 2 in 5, or over 40% or just put 41% requiring cause-specific mortality data.

• Lines 37, 40, 41, 43, 46, 49 and 54, the comma should be placed after the reference and not before to make it reader friendly. Do same across the entire manuscript.

• Line 62: include the total sample population of township of which 42 were selected. It is not very clear how this sample of 42 were selected (sample methodology).

• Since the defined sample age is 12+, it would be good to know how this age group is distributed demographically as the VA is representing this exact population especially in the study population/sample population.

• Line 137: The study title relates to ‘’adults’’ with a cut off of 12+ years and so why include under-five mortality VA data? Note in line 127, stillbirths data was collected but removed from the dataset. Hence if children below 12years VA data was collected, then it should be clearly stated that this dataset were cleaned out prior to analysis for consistency. Another issue is about what the legal definition of ‘’adult’’ is in Myanmar as 12+ years is being used as ‘’adults’’ for ease of standardization and country by country comparison? If this will pose a problem, then the title can include VA among people ‘’12 years and above’’ can be used in place of ‘’adult’’.

• Please provide reference for the statement from lines 268 to 272.

• Lines 300-303: rephrase the limitation because the objective of this study was about utility and feasibility of VA amongst 12+ and so making refences to neonates confuses the issues being studied. This clearly impacts on aspects of the conclusions in this study (line 329).

• Why didn’t AIDS show in Table 4 for females but in figure 4a it is reflected as 5% among females 12-49yrs?

Reviewer #2: This article is a meaningful thesis that systematically introduces methods that can be applied in developing countries that lack population registration systems and health personnel.

The only weakness is the lack of information about the IRB.

Nevertheless, by introducing a detailed data collection method, the verbal autopsy method suggests the possibility of supplementing autograph statistics in developing countries that lack statistical systems.

6. PLOS authors have the option to publish the peer review history of their article (what does this mean?). If published, this will include your full peer review and any attached files.

**Do you want your identity to be public for this peer review?** For information about this choice, including consent withdrawal, please see our Privacy Policy.

Reviewer #1: No

Reviewer #2: No

---

## [Decision Letter · Decision Letter 1]

5 Oct 2023

Estimating causes of community death of adults in Myanmar from a nationwide population sample: application of verbal autopsy

PGPH-D-23-01258R1

Dear Adair,

We are pleased to inform you that your manuscript 'Estimating causes of community death of adults in Myanmar from a nationwide population sample: application of verbal autopsy' has been provisionally accepted for publication in PLOS Global Public Health.

Best regards,

Collins Otieno Asweto, PhD

Academic Editor

Reviewer's Responses to Questions

**Comments to the Author**

1. If the authors have adequately addressed your comments raised in a previous round of review and you feel that this manuscript is now acceptable for publication, you may indicate that here to bypass the “Comments to the Author” section, enter your conflict of interest statement in the “Confidential to Editor” section, and submit your "Accept" recommendation.

Reviewer #1: All comments have been addressed

Reviewer #3: All comments have been addressed

Reviewer #4: All comments have been addressed

2. Does this manuscript meet PLOS Global Public Health’s publication criteria? Is the manuscript technically sound, and do the data support the conclusions? The manuscript must describe methodologically and ethically rigorous research with conclusions that are appropriately drawn based on the data presented.

Reviewer #1: Yes

Reviewer #3: Yes

Reviewer #4: Yes

3. Has the statistical analysis been performed appropriately and rigorously?

Reviewer #1: Yes

Reviewer #3: Yes

Reviewer #4: Yes

4. Have the authors made all data underlying the findings in their manuscript fully available (please refer to the Data Availability Statement at the start of the manuscript PDF file)?

Reviewer #1: Yes

Reviewer #3: Yes

Reviewer #4: Yes

5. Is the manuscript presented in an intelligible fashion and written in standard English?

Reviewer #1: Yes

Reviewer #3: Yes

Reviewer #4: Yes

6. Review Comments to the Author

Reviewer #1: Congratulations to the research team.

Reviewer #3: The authors’ idea was inspired to address such a critical issue pertaining to Myanmar’s public health and health system. The methods section is very well-written and the authors were successful in reporting their results clearly and recognizing the limitations pertaining to their study. Only minor revisions to be made.

1) In the “introduction” section, presenting similar figures from other countries would help set the scene even more, highlighting the issue of Myanmar’s statistics even further

2) How did the authors choose the methods that they implemented? Did they follow standard practice (if there is indeed standard practice, i.e., someone has conducted similar studies in the past, it should be mentioned) or published literature (again, need to explain). This is the only thing I found missing, the reasoning behind some of the choices in the study design.

3) The opening sentence of the “Leading causes of death” sub-section could be structured in a clearer way syntax-wise.

Reviewer #4: The paper represents quite informative data and analysis of community deaths in Myanmar which could also be a good replicative model for other parts of the world.

Although the reason for choosing 12 years age for adults has been mentioned and clarified that it is based on WHO VA standards to capture younger maternal deaths, ‘adult’ in this study especially in title seems to be a slight misnomer. As the study findings also conclude that cirrhosis is more common than the GDB and is found in younger age population, this would also raise a question over percentage of population with alcoholism and other addictions among teenagers. It would be interesting to see such findings in concurrence with conclusions. This would also entail specific action points for policy making.

However, this study is the most useful empirical analysis that can be carried on for policy level decisions. Appreciate the authors and sponsors of this study.

7. PLOS authors have the option to publish the peer review history of their article (what does this mean?). If published, this will include your full peer review and any attached files.

**Do you want your identity to be public for this peer review?** For information about this choice, including consent withdrawal, please see our Privacy Policy.

Reviewer #1: **Yes: **Richard AMENYAH

Reviewer #3: **Yes: **Elvira M.M. Gkrinia

Reviewer #4: **Yes: **Madhuri Devaraju
